# TORC1 Signaling in Fungi: From Yeasts to Filamentous Fungi

**DOI:** 10.3390/microorganisms11010218

**Published:** 2023-01-15

**Authors:** Yuhua Wang, Xi Zheng, Guohong Li, Xin Wang

**Affiliations:** State Key Laboratory for Conservation and Utilization of Bio-Resources in Yunnan, Yunnan University, Kunming 650091, China

**Keywords:** TORC1, Rag-GTPase, yeast, filamentous fungi, amino acids, growth, metabolism

## Abstract

Target of rapamycin complex 1 (TORC1) is an important regulator of various signaling pathways. It can control cell growth and development by integrating multiple signals from amino acids, glucose, phosphate, growth factors, pressure, oxidation, and so on. In recent years, it has been reported that TORC1 is of great significance in regulating cytotoxicity, morphology, protein synthesis and degradation, nutrient absorption, and metabolism. In this review, we mainly discuss the upstream and downstream signaling pathways of TORC1 to reveal its role in fungi.

## 1. Introduction

Target of rapamycin (TOR) is a highly conserved serine/threonine kinase that belongs to the phosphatidyl inositol (PI) 3-kinase-related kinase (PIKK) family, and serves as an important central hub of nutrient and energy signaling pathways [1]. It was initially detected in the process of screening *Saccharomyces cerevisiae* resistant strains under rapamycin treatment [2]. After TOR was identified in yeast, it was also identified in the genomes of other fungi, drosophilids, plants, mammals, and other eukaryotes [3]. Among these organisms, TORs have a similar domain structure, containing HEAT repeats (named for huntingtin, elongation factor 3, a subunit of PP2A, and TOR1) in the N-terminal region, FAT domains (named for FKBP12 Rapamycin Associated Protein (FRAP), ATM, and TRRAP) in the C-terminal region, kinase domains, and FKBP–rapamycin binding domains [4,5].

In yeasts and mammals, TOR exists in two functionally distinct complexes, rapamycin-sensitive TOR complex 1 (TORC1) and rapamycin-insensitive TOR complex 2 (TORC2) [3]. However, so far only TORC1 has been detected in microalgae and other photosynthetic eukaryotes [6]. In yeasts, TORC1 contributes to protein synthesis and cell size [7], whereas TORC2 mostly controls polarization or organization [8]. The amino acid sequence of TORC1 is highly conserved among species [9], indicating that the TORC1 of different species may have similar properties and protein substrates. However, different organisms contain relatively distinct TOR-complex components. For example, yeast TORC1 is classically composed of TOR1 or TOR2, Kog1 (RAPTOR), and Lst8; the key components of TORC1 have also been detected in the genome of *Fusarium oxysporum*, here including TOR, RAPTOR, and Lst8 [10]; while mammalian ortholog TOR complex1 (mTORC1) contains mTOR, mLST8, and RAPTOR [11].

In fungi, the upstream signals regulating TORC1 activity mainly include amino acids, carbon sources, phosphate and other nutrients, growth factors, energy, and stress factors. Nutrients, especially amino acids, directly or indirectly act on effectors of TORC1 after entering cells, while growth factors, energy, and other environmental pressures act on the TORC1 signaling pathway via unknown pathways [12]. In addition, many TORC1 regulons and substrate homologues have been logged in eukaryotic genomes [13], including leucyl-tRNA synthetase (LeuRS), adenosine monophosphate-activated protein kinase (AMPK), rat sarcoma (Ras) -related GTP-binding protein (Rag), tuberous sclerosis complex (TSC), guanine–nucleotide exchange factor (GEF), and so on. Together with these regulons, fungal cells can fine-tune their growth and development programs in response to environmental signals [14]. In many studies, TORC1 has been shown to regulate cell growth and metabolism in many aspects, mainly reflecting its regulatory role in protein synthesis, toxicity, cell morphology, phagocytosis, stress response, and other physiological processes [4].

The majority of studies on TORC1 are undertaken on model yeasts and mammals. At present, the exploration of TORC1 in other filamentous fungi also has made some progress. In order to better understand the complicated interactions of TORC1 in fungi, in this study, we reviewed stimulated signals of TORC1 and summarized its functional mode in and regulation of growth and metabolism of yeasts and filamentous fungi.

## 2. Operational Mode of TORC1 in Yeast Fungi

The TORC1 pathway is primarily stimulated by amino acids. Rag-GTPases (Rag A/C, Rag A/D, Rag B/C, and Rag B/D) figure prominently in amino acid-induced TORC1 activation of the lysosome. Budding yeast *S. cerevisiae* Rag-GTPase is a heterodimer composed of Gtr1 and Gtr2, which are highly homologous to Rag A/B and Rag C/D, respectively. TORC1 is active only when Gtr1 or RAG A/B is bound to guanosine 5′-triphosphate (GTP), and Gtr2 or RAG C/D is bound to guanosine diphosphate (GDP). Gtr1–Gtr2 is recruited to the vacuolar membrane tethering the ‘escape from rapamycin-induced growth arrest’ (EGO) ternary complex (Ego1–Ego2–Ego3) in budding yeast *S. cerevisiae* and relays amino acid signals to TORC1 [15]. The Gtr1–Gtr2 heterodimer is able to stimulate TORC1 activity via Kog1 in the presence of amino acids [16]. Additionally, analysis has shown that Kog1 affects carbon flux apportioning between glutamate/amino acid biosynthesis and gluconeogenesis through the SNF1-dependent pathway in budding yeast [17].

The Gtr1–Gtr2-dependent stimulation state of TORC1 is maintained by an intricate interplay between distinct guanine-nucleotide exchange factor (GEF) and GTPase-activating protein (GAP) complexes (Table 1) [18]. Mammalian Ragulator, besides serving as a scaffold for RAGs to regulate the localization of Rags and mTORC1 on lysosomes, has GEF activity toward RAGA/B, while its budding yeast ortholog EGO ternary complex has little GEF activity. Instead, Vam6 [19], a component of the homotypic fusion and protein sorting (HOPS) complex involved in vacuolar fusion, is regarded as a GEF that acts on Gtr1^GTP^ to positively regulate TORC1. In addition to Vam6, another complex with GAP activity for Gtr1^GTP^ that negatively affects TORC1 activity is the Seh1-associated complex (SEAC). In budding yeast, SEAC is divided into two subcomplexes. One is the Seh1-associated subcomplex inhibiting TORC1 (SEAIT), consisting of Npr2, Npr3, and the catalytic subunit Iml1, which negatively regulates TORC1 activity. The other is the Seh1-associated subcomplex activating TORC1 (SEACT), consisting of Seh1, Sec13, Sea2, Sea3, and Sea4, which can antagonize the GAP activity of SEAIT to participate in TORC1 activation [20]. SEAIT and SEACT are the relative orthologs of GAP activity toward the RAGs (GATOR) complexes GATOR1 and GATOR2 in mammals [21]. The latest research shows that the fission yeast *Schizosaccharomyces pombe* GATOR complex is necessary for TORC1 regulation in case of amino acid deficiency, as it can inhibit TORC1 by activating the general control nonderepressible kinase 2 (GCn2) pathway and induce autophagy [22]. In addition to the above-mentioned complexes, recently it was reported that Lst4–Lst7 (homologous to mammalian FLCN–FNIP) with GAP activity acts on Gtr2^GDP^, contributing to the formation of Gtr1^GTP^–Gtr2^GDP^, which results in TORC1 activation [23].

TORC1 deactivation is an active process induced by amino acid starvation. It has been reported that inactive Gtr1 binding to GDP results in TORC1 inhibition by means of the TORC1 component Tco89 [24]. In addition, some negative regulators of TORC1 have been detected in budding yeast *S. cerevisiae* (Table 1). For example, in case of amino acid starvation, G-protein-coupled receptor (GPCR)-like protein Ait1 inhibits TORC1 via Gtr1/2 using a loop that resembles the Rag A/C binding domain in the human SLC38A protein [25]. Ivy1 is a protein containing the I-BAR domain, which binds to the vacuolar membrane and interacts with the EGO complex to inhibit Gtr-induced TORC1 activation [26]. Some studies reported that yeast Whi2 [27], ubiquitin [28], and increased free uncharged tRNA [29] also inactivate TORC1. It has also been reported that the activation of AMPK can produce an inhibition effect on TORC1 when the cells encounter unfavorable conditions such as energy deficiency, stress, malnutrition, hypoxia, or DNA damage [30].

Recently, related reports indicated that there is an independent sensing mechanism specifically for perceiving glutamine, in addition to the Rag/Gtr complex [31]. In the *S. cerevisiae* model, the phosphoinositide (PI)-3 kinase complex Vps34–Vps15 and protein Pib2 containing a FYVE domain have been reported to participate in the sensing mechanism of glutamine (Table 1). The role of Vps34–Vps15 in the TORC1 reaction to glutamine is to recruit Pib2 to the vacuolar membrane. Pib2 bound to the vacuolar membrane through its FYVE domain is targeted to TORC1 by binding to Kog1 via its middle portion. Strikingly, Pib2 has two antagonistic domains against TORC1, an N-terminal TORC1 suppression domain and a C-terminal TORC1 activation domain. Still, how the antagonistic activity of pib2 influences TORC1 activity remains to be verified [32].

## 3. Upstream Signals of TORC1 in Yeast Fungi

### 3.1. Amino Acid Signaling of TORC1 in Yeast Fungi

Nitrogen is essential for the synthesis of amino acids, nucleotides, and other cellular components. Exogenous amino acids are important nitrogen sources, the quality of which plays an important role in the mechanism of amino acid-induced TORC1 activation in yeasts. Glutamine and ammonium are considered the best nitrogen sources in a Pib2-dependent manner, not in the Rag-GTPase regulation model [31]. The Vps34–Vps15 complex and protein Pib2 are involved in glutamine-induced activation of TORC1 [33]. Leucine, methionine, and perhaps other amino acids effectively stimulate TORC1 via the GTPases Gtr1 and Gtr2. Leucine binding to leucine-tRNA synthetase (LeuRS /Cdc60) acts on Gtr1^GTP^ to block GTP hydrolysis, thereby regulating the activity of TORC1. In contrast, mammalian leucine binding to LeuRS only acts on Rag D rather than Rag A/B/C [34]. Additionally, when the level of methionine in the cytoplasm is high enough, its C-terminus is methylated to form S-adenosylmethionine (SAM), thus the catalytic subunit of PP2A (PP2Ac) is methylated to resist the assembly of SEACIT (Figure 1) [11].

### 3.2. Carbon Signaling of TORC1 in Yeast Fungi

Glucose is one of the most basic carbon sources; it has a positive effect on maintaining the homeostasis of the body. In budding yeast *S. cerevisiae*, on the one hand, glucose induces the activation of TORC1 in both a Gtrs-dependent and Gtrs-independent manner in case of glucose sufficiency [35]. On the other hand, TORC1 is deactivated without Gtr1/2 under the condition of glucose deprivation. Downregulation of TORC1 activity in response to glucose starvation in yeast partially hinges on Snf1/AMPK kinase [36]. Activated Snf1 phosphorylates Kog1, which is located in nearby Ser491 and Ser494, leading to Kog1 being separated from TORC1. This leads to lower activation of TORC1 (Figure 2) [37]. In fact, the pathway inactivates TORC1 to ensure that cells stay in the quiescent stage and survive under starvation conditions by increasing its activation threshold [38]. Recently, a new factor, Ecm33, was reported to be involved in the glucose uptake of yeast TORC1 signals, which may activate TORC1 by regulating the activity of Snf1 or protein kinase A (PKA) [39]. In addition, it was also reported that glucose regulates TORC1 activity by changing the pH of the cytoplasm, promoting a physical interaction between v-ATPase and Gtr1 [40]. However, TORC1 activity is required for establishing and maintaining glucose transcriptional levels. TORC1 regulates the transcriptional response to glucose and the developmental cycle via the Tap42–Sit4–Rrd1/2 pathway in *S. cerevisiae* [35]. In addition to glucose, fatty acids, glycerol, and carbon dioxide can also be used as carbon sources by fungi, but how these carbon sources connect to TORC1 still needs to be explored. Glycerol and glucose are the preferred carbon sources for *Yarrowia lipolytica*, but its morphology differs depending on the two carbon sources [41,42].

### 3.3. Phosphate Signaling of TORC1 in Yeast Fungi

Phosphate is the third nutrient regulated by the TOR signaling pathway. Liu et al. detected that *Candida albicans* can sense changes in intracellular phosphate concentrations via the phosphate transporter Pho84 to regulate the activity of TORC1 [43]. In order to cope with variable and scarce phosphate conditions, *S. cerevisiae* has evolved a phosphate-response signaling and metabolic pathway, called the phosphate homeostasis (PHO) pathway, for monitoring phosphate cytoplasmic levels [44]. Pho84 is a phosphate transporter with high affinity and is upregulated during phosphate starvation events. Research on how phosphate affects cell growth and metabolism identified a link between phosphate signaling and the PKA pathway in *S. cerevisiae*. The addition of phosphate in the presence of glucose has been shown to upregulate the PKA pathway and causes trehalose activation, trehalose mobilization, and suppression of stress response element (STRE) genes [45]. In addition, in the *S. cerevisiae* model, Pho84 can also regulate Gtr1-mediated TORC1 activation, and the state of the intracellular phosphate pool and extracellular phosphate concentration are integrated to regulate phosphate acquisition; in turn, TORC1 can also provide feedback on the PHO regulon (Figure 3).

In addition to carbon sources, nitrogen sources, and phosphorus, stressors such as growth factors, oxygen, osmotic pressure, high temperature, and caffeine can also affect the activity of TORC1 [46]. However, we know little about the sensing mechanism and regulatory factors.

## 4. Effects on Fungal Growth and Metabolism

TORC1 regulates different downstream factors to produce diverse signaling pathways, which are integral to regulating growth and development in different fungi. TORC1 directly interacts with an extensive array of kinases and phosphatases in fungi to control protein synthesis, cell morphology, toxicity, mycelial growth, nitrogen metabolism, and other physiological processes (Table 2). In fungi, TORC1 mainly regulates cell growth and metabolism through two branches, Sch9 and Tap42.

### 4.1. Modulation of Protein Synthesis by TORC1

Controlling protein synthesis is one of the key functions of TORC1. Recent studies have shown that TORC1 controls protein synthesis by regulating ribosome biosynthesis, protein translation, and other processes. In coordination, the yeast TORC1 signaling pathway and the general amino acid-control (GAAC) signaling pathway regulate protein kinase GCN2 activity. The conservative GAAC signaling pathway coordinates the availability of amino acids with the initiation of translation to enable cells to adapt to amino acid starvation. In amino acid-deficient cells, uncharged tRNA binds and activates eIF-2-alpha kinase GCN2 [47]. Dephosphorylated Tap42-PPase in the TORC1 signaling pathway can increase the activity of GCN2 [48]. Activated GCN2 induces the phosphorylation of eIF2α, ultimately leading to the inhibition of mRNA translation [49]. In addition to reducing the synthesis of most proteins in vivo, phosphorylation of eIF2α also regulates the expression of other genes via transcription factors, such as yeast transcription factor Gcn4 and mammalian ATF4, which can induce the expression of amino acid transporters, enzymes involved in amino acid metabolism, and factors associated with autophagy to adapt to amino acid deficiency [11]. In addition, in *D. discoideum*, knocking out DdTOR in strains using gene silencing modulates 4EBP1 phosphorylation and reduces its association with eIF4E, lowering translation inhibition. TORC1 regulates the overall level of translation to control protein synthesis, which controls cell growth and life extension.

Ribosomes are indispensable organelles that ensure normal protein synthesis, which has an important effect on fungal cell growth. Based on data from previous studies, the ribosome biogenesis (Ribi) regulon, the ribosome protein (RP), ribosomal RNA, and tRNA are the master factors regulating ribosome biogenesis, and the transcription of these factors depends on the activation of RNA polymerase I, II, and III [50]. In *S. cerevisiae*, TORC1 requires the mediation of Sch9 for regulating ribosome biogenesis. Recently, another AGC kinase, serine/threonine-protein kinase ypk3, which is highly homologous to human S6 kinase (S6K), was also reported to be a key downstream component of the TORC1 pathway, and is required for phosphorylation of Rps6 (ribosomal protein S6) at Ser-232 and Ser-233 [51]. tRNA and 5S rRNA are the products of RNA pol III. Sch9 activates RNA pol III to promote gene expression of 5S rRNA and tRNA by inhibiting phosphorylation of the repressor of RNA polymerase III transcription MAF1 homolog [52]. Ribosome proteins are mainly synthesized via RNA pol II, and the transcription of RNA pol II is activated by the transcription factor Ifh1 [53], but also partly depends on Sch9. TORC1 also stimulates RNA pol I gene expression of rDNA by regulating the transcription initiator Rrn3 in a Sch9-dependent and -independent manner. The transcriptional repressors Stb3, Dot6, and Tod6 contribute to inhibiting gene expression of the RP and Ribi regulon [50]. Dot6 and Tod6 regulate the Ribi regulon to affect cell size, while Stb3 regulates RP gene expression to affect the cell growth rate. The TORC1 downstream factor Sch9 can inhibit phosphorylation of Stb3, Dot6, and Tod6, thereby promoting gene expression of RP and Ribi regulon. In *S. pombe*, ribosome biosynthesis also depends on the TORC1 signaling pathway, but differs from *S. cerevisiae* in some regulatory targets. For example, in *S. pombe*, phosphorylation of Maf1 is directly regulated by TORC1 [54]. Additionally, in *S. pombe*, three AGC kinase family members, Psk1, Sck1, and Sck2, are highly homologous to human S6k1, which is directly phosphorylated by TORC1 [55]. Among these, Psk1 is vital to phosphorylation of Rps6 [55].

### 4.2. Modulation of Metabolic Programs by TORC1

Fungal cells can sense nitrogen-containing nutrients in the environment and can adjust their transcription, metabolic synthesis, and other life processes. The TORC1 signaling pathway is necessary for sensing, transport, and catabolic regulation of nitrogen sources. The Ssy1–Ptr3–Ssy5 (SPS) pathway modulates efficient amino acid uptake by responding to extracellular amino acids and only exists in fungi. The primary amino acid sensor is a plasma membrane localization complex (called SPS sensor) composed of Ssy1, Ptr3, and Ssy5 [56], which is related to genes involved in amino acid permease expression [57]. TORC1 enhances the stability of nuclear Stp1, the key regulon of the SPS pathway, promoting amino acid uptake through phosphatase (PP2A or Sit4) in *S. cerevisiae* [58].

The nitrogen catabolism-repression (NCR) signaling pathway is closely connected to the transcription level of nitrogen catabolism, which responds to nitrogen metabolism with the mediation of TORC1. In *S. cerevisiae,* the regulation of NCR gene expression involves four transcription factors: the GATA-like transactivators Gln3 and Gat1, and the GATA-like transinhibitors Gzf3 and Dal80 [59]. Gln3 and Gat1 are closely connected to the expression of inhibitory genes of nitrogen catabolism and accumulation of lipids [60,61]. In the condition of a sufficient nitrogen source, the Tap42–PP2A and Tap42–Sit4 complexes prevent dephosphorylation of Gln3 and Gat1 [62], which hinders their transfer to the nucleus. Dephosphorylated TORC1 impedes the combination of Tap42 and PP2A (sit4), and free Tap42–PP2A is beneficial for the activation of the downstream transcription factors Gat1 and Gln3. In addition to the above-mentioned transcription factors, some transcription regulators are required to regulate certain genes in specific metabolic pathways of certain nitrogen sources. For example, Aro80 and Dal81 are required for the gene transcription of other nitrogenous compounds, such as aminobutyric acid [63]. In addition, in *S. pombe*, TORC1 promotes the phosphorylation of GATA transcription factor Gaf1 to control nitrogen response genes by inhibiting the PP2A-like phosphatase Ppe1 [64].

In addition to the role of nitrogen metabolism, the TORC1 downstream factor Sch9 is important for sphingolipid metabolism. It adjusts sphingolipid metabolite levels according to cell needs by controlling the gene expression of Ydc1 and Ypc1 ceramidase in *S. cerevisiae* [65].

### 4.3. Modulation of Macroautophagy by TORC1

Macroautophagy is the process of degradation of cytoplasmic proteins and organelles via the lysosomal pathway, which is controlled by an autophagy-related protein (Atg). In *S. cerevisiae*, TORC1 can indirectly or directly regulate autophagy. First, TORC1 directly inhibits Atg13 phosphorylation to prevent assembly of the Atg1–Atg13 initiation complex [66]. Moreover, a study of *S. pombe* reported that phosphorylation of Atg13 relies on TORC1 as it does in *S. cerevisiae* [67]. Second, under conditions of nutrient starvation, inactivation of Sch9 leads to induction of autophagy, which requires Atg1 protein kinase, Rim15, and Msn2/4 transcription factors. An additional mechanism involving the Tap42–PP2A complex may also affect macroautophagy [52]. In addition, in *S. cerevisiae*, cells lacking Ecm33 mediate the dephosphorylation of the TORC1 substrates Atg13 and Sch9 to induce autophagy [39].

### 4.4. Modulation of Stress Response by TORC1

Under stress conditions, cells need to adjust the transcription levels of stress response genes to maintain homeostasis. Msn2/4 and Gis1 are important transcription factors for regulating the expression of stress response genes. In *S. cerevisiae*, TORC1 downregulates the activity of Yak1 and Rim15 kinases via Sch9 to modulate the phosphorylation of Msn2/4 and Gis1 in response to various nutritional starvation conditions [68]. Besides its role in transcriptional activation, Rim15 also takes part in post-transcriptional mRNA stabilization to avoid 5′–3′ degradation of mRNA [69]. In case of stress, Msn2p/Msn4p can also promote gene expression of UGP1 to regulate the glycometabolism. UGP1, uridine diphosphate (UDP)-glucose pyrophosphorylase, is a key catalytic enzyme for the production of UDP glucose (UDPG), which plays a key role in the carbohydrate metabolic pathway [70]. Additionally, Yak1 activates the heat-shock transcription factor 1 (Hsf1) during glucose starvation [71].

Recently, Liang et al. explained that the reason why the TORC1–Sch9 signaling pathway inhibits the yeast-to-hypha dimorphic transition in the presence of glycerol in *Y. lipolytic* is that the transcription of Rim15 is inhibited. The TORC1–Sch9–Rim15 signaling pathway is very important for the yeast-to-hypha dimorphic transition [72]. In addition to the mentioned stress factors, in *C. auris*, it was observed that a kind of lncRNA, DINOR, also regulates a series of stress reactions, and the function of DINOR are connected to TOR signaling [73].

### 4.5. Modulation of Pathogenicity and Virulence by TORC1

The TORC1 signaling pathway also affects the growth and virulence of pathogenic fungi, but many of the regulatory mechanisms are still unclear. In order to better understand the signaling pathway of TORC1 in pathogenic fungi, we need to find its upstream and downstream regulatory factors at a deeper level. Clarifying the mechanism of TORC1 in animal and human pathogens will be helpful in providing new schemes for disease treatment.

*M. oryzae* is a kind of pathogenic fungus that causes destructive disease in rice. Among the products downstream of TORC1, the PP6 catalytic subunit MoPpe1, which interacts with MoTap42, takes part in the development [74], toxicity, and activation of the CWI pathway. Related studies identified and characterized a Tap42 interacting protein 41 (MoTip41), which interacts with MoPpe1 and participates in the crosstalk between the TOR and CWI signaling pathways during *M. oryzae* infection [75]. In addition, it has become known that autophagy is necessary for the formation of functional appressoria and pathogenicity and is an important guarantee for the effective defense against *M. oryzae* [76]. Marroquin-Guzman et al. detected that MoAtg3 and MoAtg9 affect the processes of autophagy, glycogen transfer, and lipid degradation, indirectly leading to decreased pathogenicity of appressoria, which may be related to abnormal regulation of the TORC1 pathway and autophagy-related pathways that control appressoria function and infection growth [77]. Research on *F. oxysporum* detected those amino acids, especially cysteine, can significantly affect the growth of hyphae and the synthesis of T-2 toxin, which is regulated through the Gtr/Tap42 pathway [78]. In addition, *F. oxysporum* TOR is also necessary for ammonium-mediated inhibition of invasion and growth [79].

*C. orbiculare* is an anthracnose pathogen in Cucurbitaceae plants. TORC1 activity is regulated by CoWhi2 when forming the infectious appressorium structure. CoMsn2 is a homologue of Msn2p. During periods of nutrient deficiency, it can cause related gene expression in the infection stage mediated by CoMsn2 [80]. In *C. neoformans*, VAD1, a major determinant of virulence, is regulated by TORC1 [81]. Current research has confirmed that the VAD1–EM15–CAC2 regulatory pathway is a component of starvation/TOR virulence response through the sugar transporter gene STL1 [82]. At the same time, the cyclic adenosine monophosphate (cAMP) nutrient sensing pathway affects the pathogenesis of *C. neoformans* [83], and there is also a potential link between TORC1 and PKA. In *A. fumigatus*, the HOG pathway and the calcineurin/CrzA pathway have important effects on the stress response and toxicity of mammalian hosts, regulated by SchA, which is homologous to Sch9 of *S. cerevisiae* [84]. In addition, *A. fumigatus* TOR, a repressor of the ornithine biosynthesis gene, takes part in the production of an iron carrier [85].

In addition to the regulation of pathogenicity and virulence in pathogenic fungi, TORC1 also regulates the cell cycle. For example, in *C. auris,* TOR inhibition leads to G1 cell cycle arrest and defective cytokinesis [86]. In *U. maydis*, the TORC1–Sch9 signaling pathway inhibits cyclin-dependent kinase 1 (Cdk1) and increases the activity of Wee1 kinase by the activation of PP2A-B55, causing cells to enter the G2 phase [87]. Additionally, the Ras-like GTPases (Don1, Don3, and Cdc42) participate in the process of diaphragm formation [88].

### 4.6. Modulation of Ubiquitination by TORC1

Ubiquitination refers to the process by which ubiquitin specifically modifies target proteins in the action of a range of enzymes. It makes a difference in the regulation of protein stability, localization, activity, and interaction. Gap1 is an amino acid transporter with a wide range of substrates and high substrate affinity. It regulates the abundance of amino acids in cells through oligoubiquitin- and polyubiquitin-induced endocytosis [89]. In *S. cerevisiae*, TORC1 directly interacts with Npr1 kinase to regulate the growth and development of cells. Npr1 kinase, a negative regulator of endocytosis, is responsible for stabilizing Gap1p on the plasma membrane, and for vacuolizing and degrading high-affinity permeates under starvation conditions. Activated TORC1 inhibits Npr1 and targets the PP2A-type Sit4 phosphatase to the functionally redundant arrestin-like Bul1 and Bul2 proteins. Bul1 and Bul2 proteins are then dephosphorylated and dissociate from the 14–3–3 proteins [89], leading to rsp5-mediated ubiquitination of Gap1 [90].

Mie2, an RNA binding protein in *S. pombe*, plays an important role in the regulation of sexual differentiation and meiosis [91]. TORC1-induced Mei2 polyubiquitination and proteasome degradation lead to the inhibition of sexual differentiation. Another transcription factor, Ste11, has been reported to control differentiation, which requires phosphorylation of TORC1 and Mie2 [92]. A new study recently detected that TORC1 is also part of the ubiquitin proteasome system, participating in cell proliferation and gametogenesis in *S. pombe*, which provides a strategy for the treatment of human diseases [93].

## 5. Conclusions

TORC1 can work with various environmental signals, thus regulating cell growth and development. It is known that growth factors stimulate the small GTPase Ras-homolog enriched in brain (Rheb) to bind and activate mTORC1 by the PI3K–PDK1–AKT signaling pathway in mammals [11], while in *S. cerevisiae*, it is not known whether Rhb1 affects TORC1 activity, even though it has been reported that the function of Rhb1 resembles Rheb [94]. In addition to glutamine, leucine, and methionine, we should prove how cells regulate TORC1 activity in the presence of other amino acids like arginine and whether other amino acids also depend on Gtrs in the fungi kingdom. TORC1 is regulated by nutrients and environmental stress factors, but it is still unknown whether there are other models apart from the above-mentioned model. A study on mammals shows that Rag GTPases may have positive and negative regulatory effects on TORC1 activity, yet it is not known whether there is the same effect in filamentous fungi and yeasts [94]. Additionally, few studies on fungi refer to the related TORC1 growth factor complex or environmental stress signaling pathways, so how these signals connect to TORC1 still needs to be validated. Therefore, how TORC1 activity is changed in response to environmental signals upstream of the TORC1 signaling network, is still one of the important concerns and demands further exploration of the TORC1 signaling pathway.

Downstream of the TORC1 signaling network, TORC1 radiates multiple signaling pathways. In recent years, it has been proven that TORC1 produces a cascade-amplification effect involving a variety of kinases or phosphatases. Recent phosphoproteomics studies have shown that there are some new growth-related effectors acting downstream of the TORC1 pathway, but the identification of these effectors still needs be further studied. Clarifying the downstream events of TORC1 in fungi not only has an important impact on biological control, but also plays a role in treating human diseases. For example, some biological processes such as ribosome biogenesis and phagocytosis are closely related to health of the body. Therefore, determining the characterization of the TORC1 signaling pathway will provide new strategies for the research of many target drugs. With the development of genomics, transcriptomics, proteomics, and other omics approaches and technologies, it is certain that the role of transcriptional factors, proteins, and other genes in the TORC1 functional mechanism in fungi will be revealed in the near future.

## Figures and Tables

**Figure 1 microorganisms-11-00218-f001:**
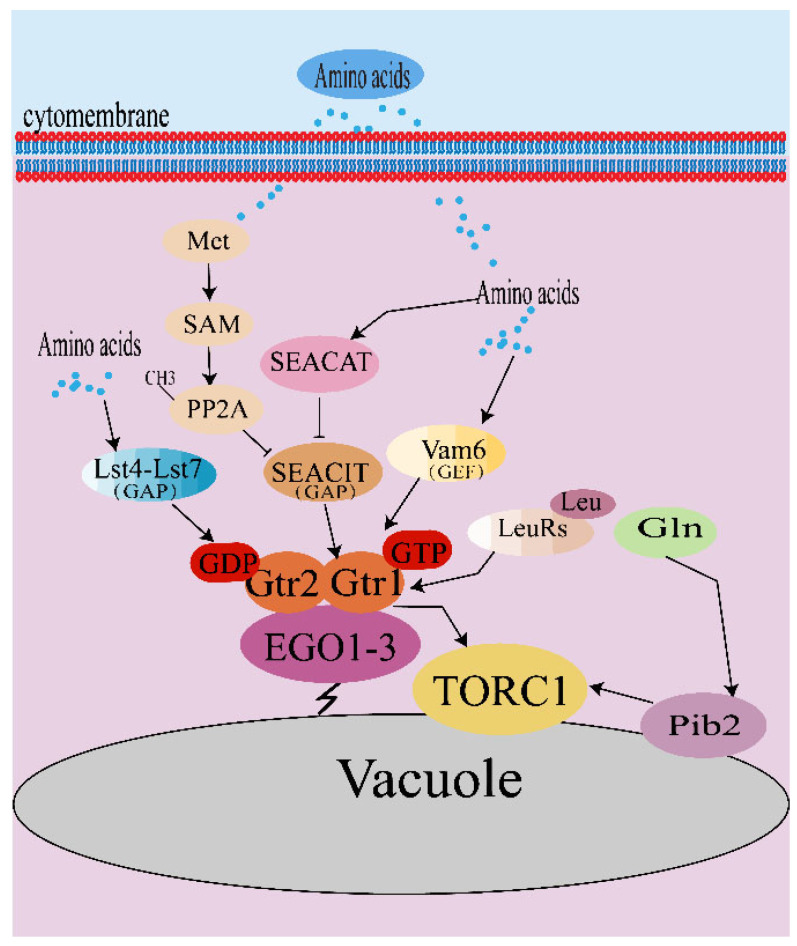
Amino acids regulate TORC1 activity in *S. cerevisiae*. TORC1 activation stimulated by amino acids mainly depends on the stimulation state of Gtrs, and the stimulation state of Gtrs is closely regulated by GEF and GAP proteins. LeuRS has also been reported to play a role in TORC1 activation. In addition, glutamine-induced TORC1 activation need the mediation of Pib2.

**Figure 2 microorganisms-11-00218-f002:**
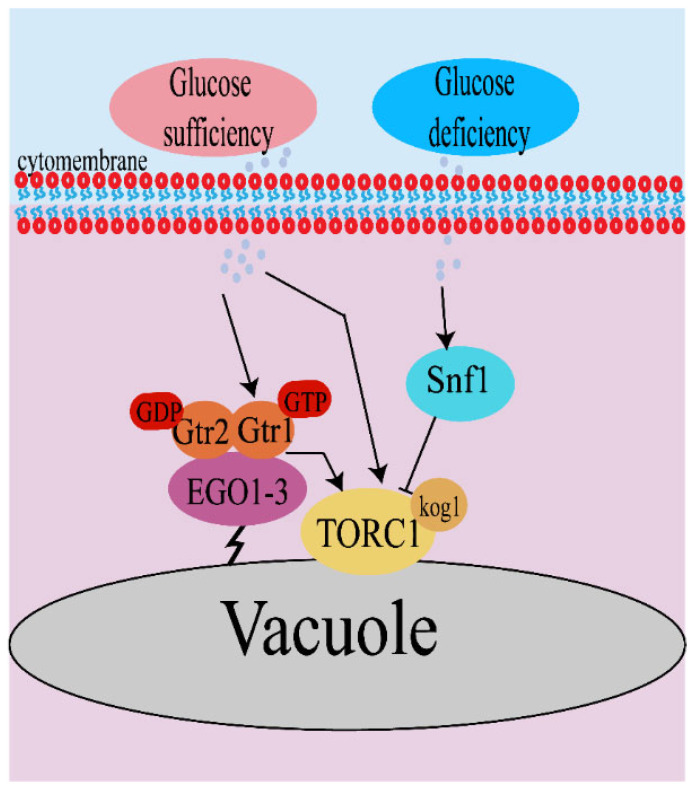
Glucose regulates TORC1 activity in S. cerevisiae. Under condition of glucose sufficiency, glucose activates TORC1 in both a Gtrs-dependent and -independent manner. Under condition of glucose deficiency, glucose can inhibit TORC1 activity via Snf1.

**Figure 3 microorganisms-11-00218-f003:**
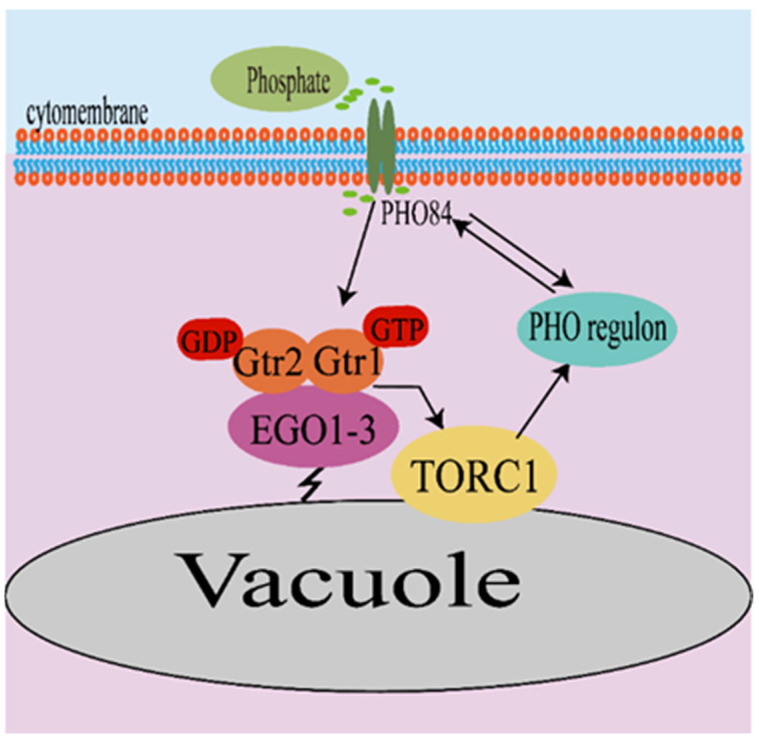
Phosphate regulates TORC1 activity in S. cerevisiae. In the S. cerevisiae model, Pho84 can sense the changes of intracellular phosphate concentrations to regulate Gtr1-mediated TORC1 activation; in turn, TORC1 also regulates the PHO regulon.

**Table 1 microorganisms-11-00218-t001:** The regulatory factors involved the operational mode of TORC1, their mammalian orthologs and functions or functional mechanisms in fungi.

Regulatory Factors	Fungal Species	Human Homolog	Function or Functional Mechanisms
Rag-GTPase (Gtr1/Gtr2)	*S. cerevisiae,* *S. pombe,* *F. oxysporum*	Rag A/B, Rag C/D	recruits TORC1 to vacuolar membrane
EGO ternary complex	*S. cerevisiae,* *S. pombe*	Ragulator	as a scaffold tethering RAG heterodimer on vacuolar membrane
Vam6	*S. cerevisiae,* *S. pombe*	Ragulator	has GEF activity that acts on Gtr1 ^GTP^ to positively regulate TORC1
Seh1-associated complex (SEAC) (SEAIT and SEACT)	*S. cerevisiae* *S. pombe* (GATOR complex)	GATOR complex (GATOR1 and GATOR2)	SEAIT/GATOR1: has GAP activity that acts on Gtr1 ^GTP^ to negatively regulate TORC1 SEACT/GATOR2: antagonizes the GAP activity of SEAIT to participate in the activation of TORC1
Lst4–Lst7	*S. cerevisiae* *S. pombe* (Lst4– Bhd1)	FLCN–FNIP	acts as GAP of Gtr2 to stimulate the activity of TORC1
Vps34–Vps15	*S. cerevisiae*	Vps34–Vps15	recruits Pib2 to the vacuolar membrane in TORC1 glutamine reaction
Pib2	*S. cerevisiae* *S. pombe* (SPBC9B6.03 protein)	-	bound to the vacuolar membrane through its FYVE domain is targeted to TORC1
Ait1	*S. cerevisiae*	-	binds to the vacuolar membrane and interacts with EGO complex to inhibit Gtr-induced TORC1 activation
Ivy1	*S. cerevisiae*	missing-in-metastasis (MIM)	a negative regulator of Gtr-dependent-TORC1 activation
Whi2	*S. cerevisiae*	potassium channel tetramerization domain containing 11 (KCTD11) protein	a conserved negative regulator of TORC1 in response to low amino acids
Free uncharged tRNA	*S. cerevisiae* *S. pombe*	tRNA	deactivates TORC1
Ubiquitin	*S. cerevisiae*	Ubiquitin	regulates the degradation of Kog1 and the stability of TORC1 through non-covalent binding to Kog1

**Table 2 microorganisms-11-00218-t002:** TORC1 regulates physiological cell processes via regulatory factors.

Fungi Species	Regulatory Factors	Regulation of Physiological Processes
*S. cerevisiae*	Tap42-PPase, Gcn2, and eIF2α	protein translation
	Sch9, ypk3, MAF1, Ifh1, Stb3, Dot6, and Tod6	ribosome biosynthesis
	PP2A or Sit4, Stp1, Gln3, and Gat1	sensing, transport, and catabolic regulation of nitrogen sources
	Sch9, Ydc1, and Ypc1 ceramidase	sphingolipid metabolism
	Atg13, Rim15, and Msn2/4, Tap42-PP2A, and Ecm33	macroautophagy
	Npr1	ubiquitination of Gap1
	Yak1 and Rim15, Msn2/4, and Gis1	stress response
*S. pombe*	Gcn2 and eIF2α	protein translation
	AGC kinases and Psk1	phosphorylation of Rps6
	PP2A-like phosphatase Ppe1 and Gaf1	nitrogen metabolism
	Atg13	macroautophagy
	Mie2, Ste11	sexual differentiation
	-	cell proliferation and gametogenesis
*Y. lipolytic*	Sch9–Rim15	the yeast-to-hypha dimorphic transition
*Dictyostelium discoideum*	4EBP1 and eIF4E	protein translation
*Candida auris*	DINOR	stress response
	-	G1 cell cycle arrest and cytokinesis defect
*Magnaporthe oryzae*	PP6 catalytic subunit MoPpe1 and MoTap42	toxicity and activation of the cell-wall integrity (CWI) pathway
	MoATG3, MoAtg9, and Atg13	autophagy, control appressoria function and infection growth
*F. oxysporum*	Gtr/Tap42 pathway	the growth of hyphae and the synthesis of T-2 toxin
*Coletothrichum orbiculare*	CoWhi2 and CoMsn2	pathogenicity and virulence
*Cryptococcus neoformans*	STL1 and VAD1–EM15–CAC2	virulence
*Aspergillus fumigatus*	SchA	the regulation of high osmolarity, glycerol (HOG), and calcineurin/CrzA pathway calcineurin, as well as stress response and toxicity of mammalian host
	-	production of iron carrier
*Ustilago maydis*	Ras-like GTPases (Don1, Don3, and Cdc42)	diaphragm formation
	Sch9, PP2A-B55, and PP2A-B55	inhibit Cdk1; cytokinesis

## Data Availability

Not applicable.

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
