# Peer review of "TORC1 Signaling in Fungi: From Yeasts to Filamentous Fungi"

_microorganisms, 2023, doi:10.3390/microorganisms11010218_

Round 1

Reviewer 1 Report

this manuscript is well written but the authors need to pay attention in some issues

the major concern,

based in the information of the manuscript the authors should provide new hypothesis to be tested in the future to enrich the field

minor concerns

in figure legends. for better understandings of the figures a brief description of them will be better for the readers.

table 2. describe the abbreviation of the scientific names of the microorganisms. U. maydis? Ustilago maydis? etc. and make sure when the microorganism is first describe in the manuscript, write down its full name.

when abbreviations are used, describe their meaning, for example

4.2   “The SPS pathway “, what SSP means.  check in all manuscript.

Author Response

Response to Reviewer 1 Comments

the major concern

Point 1: based in the information of the manuscript the authors should provide new hypothesis to be tested in the future to enrich the field.

Response 1: We have supplemented the discussion part of the article and put forward some new assumptions, future prospects and potential applications about this topic. Revised portions in the page 12 are marked in blue.

the minor concerns

Point 2: in figure legends, for better understandings of the figures a brief description of them will be better for the readers.

Response 2: We have briefly described the content of figures 1, 2, and 3. Revised portions below the figures1,2,3 of page 5,6,7 are marked in blue.

Point 3:  table 2. describe the abbreviation of the scientific names of the microorganisms. U. maydis? Ustilago maydis? etc. and make sure when the microorganism is first described in the manuscript, write down its full name.

Response 3: We have revised the full name of the microorganism species that first appeared in the table2 and article. Revised portions in the page10,4.4, page11,4.5 and the first column of page 8 table2 are marked in blue.

Point 4:  when abbreviations are used, describe their meaning, for example 4.2 “The SPS pathway “, what SSP means. check in all manuscript.

Response 4: We have made supplementary explanations for the meaning of some abbreviations in the article. Revised portions in the first paragraph of page1, page 2, the first paragraph of page5,3.2, page6, 3.3, page8, 4.1, page9, 4.2, page10, 4.4, and page 11,4.5 are marked in blue.

Reviewer 2 Report

In this review, the authors mainly discuss the upstream and downstream signaling pathways of TORC1 to reveal its role in fungi. The review is well written, comprehensive and well presented.

The authors are just advised to clarify the figures 1-2-3  for the respective readers  and rephrase the figures titles to reflect and explain the figs

Also All references should be in the Journal format

Author Response

Response to Reviewer 2 Comments

Point 1: the authors are just advised to clarify the figures 1-2-3 for the respective readers and rephrase the figures titles to reflect and explain the figs. 

Response 1: We have briefly described the content of figures 1, 2, and 3, and rephrase the figures titles. Revised portions below the figures 1,2,3 of page 5,6,7 are marked in blue.

Point 2: All references should be in the Journal format.

Response 2: We have adjusted the references format in the article according to the journal reference format. Revised portions in the page12-17 references are marked in blue.